# Interpretable Machine Learning-Based Risk Scoring with Individual and Ensemble Model Selection for Clinical Decision Making

**Han Yuan & Jin Wee Lee & Mingxuan Liu & Siqi Li & Chenglin Niu**
Duke-NUS Medical School
National University of Singapore

**Jun Wen**
Faculty of Arts and Social Sciences
University of Zurich

**Feng Xie**
School of Medicine
Stanford University

## Abstract

Clinical scores are highly interpretable and widely used in clinical risk stratification. AutoScore was previously developed as a clinical score generator, integrating the interpretability of clinical scores and the discriminability of machine learning (ML). Although a basic framework has been established, AutoScore leaves room for enhancement. In this work, we improved AutoScore with additional variable ranking methods and an automatic model selection. We demonstrated that these updates generate clinical scores with fewer variables and higher accuracy. Code is open access[1].

## 1 Introduction & Method

Clinical scores, conventionally derived from expert opinions, have been widely applied in clinical risk stratification for their accessibility and interpretability (McGinley & Pearse, 2012; Xie et al., 2022c). To assist clinicians in tailoring clinical scores across various settings, Xie et al. (2020; 2022b) developed an automatic clinical score generator named AutoScore (See Figure 1).

Despite AutoScore's success in generating transparent and interpretable clinical scores for risk stratification across various clinical settings (Yuan et al., 2022; Xie et al., 2022a), its current framework still leaves room for improvements, i.e., (1) variable ranking: AutoScore applies random forest (RF) (Breiman, 2001) to rank variables, which is biased in some scenarios (Strobl et al., 2007; Nicodemus, 2011) and alternative variable ranking methods might enhance the ranking and generate better clinical scores; (2) model selection: AutoScore requires manual model selection by users and an automatic option will make AutoScore more user-friendly and consistent with the "Auto" feature.

In this study, we explored five alternative variable ranking methods for the replacement of RF. Among them, two are statistical methods (two-sample Student's t test (Student, 1908) and Wilcoxon's rank sum test (Wilcoxon, 1947)) and two are ML methods (XGBoost (Chen & Guestrin, 2016) and SHAP (Lundberg & Lee, 2017)). An ensemble ranking method of majority voting is also implemented (See Appendix Section A).

For the development of automatic model selection, we establish it based on the original framework: AutoScore prepares multiple candidate clinical scores (models) with different variables and a parsimony plot to illustrate the relationship between clinical scores' performance and complexity (variable numbers). Users then manually select the optimal clinical score (model) based on domain knowledge. In this study, after getting the parsimony plot, a "thresholding" strategy would cut in and replace the previous manual selection with an automatic method (See Appendix Section A).

---

[1] https://github.com/Han-Yuan-Med/comparison

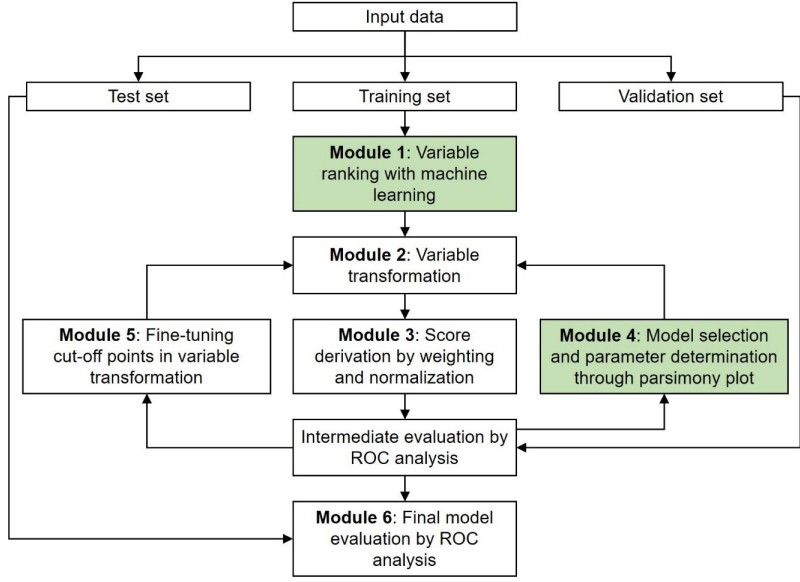

Figure 1: The pipeline of AutoScore. Modules 1 and 4 in green are upgraded in this work.

## 2 EXPERIMENTS & CONCLUSION

We used a dataset of 44,918 de-identified patients from the Beth Israel Deaconess Medical Center (BIDMC) critical care units (Johnson et al., 2016). Among them, 26,952 (60%), 8,983 (20%), and 8,983 (20%) unique patients were randomly split as the training, validation, and test set, respectively. We chose a total of 21 candidate variables for predictive modeling, including temperature, heart rate, age, etc. Missing variables were imputed using the cohort median.

We compared the clinical scores created by AutoScore with different variable ranking methods in conjunction with the automatic model selection in terms of AUROCs on the test set (95% confidence intervals (CIs) were computed using bootstrap). Table 1 shows the prediction performance of clinical scores. The RF-based baseline used 7 variables and had an AUROC of 0.746 (95% CIs 0.728-0.764). Using fewer variables, clinical scores built on the t test, Wilcoxon's rank sum test, and majority voting presented better discrimination performance, which is crucial in the fast-paced hospital environment. Also, the clinical score developed by XGBoost and SHAP yielded better AUROCs using the same amount of variables as the baseline. Other details are available in Appendix Section B.

Table 1: Prediction performance of various scores built on AutoScore with diverse variable ranking

| Ranking method | Variables | AUROC (95% CIs) |
|---|---|---|
| random forest (baseline) | 7 | 0.746 (0.728-0.764) |
| t test | 5 | 0.754 (0.737-0.772) |
| Wilcoxon's rank sum test | 6 | 0.761 (0.744-0.777) |
| XGBoost | 7 | 0.750 (0.733-0.767) |
| SHAP | 7 | 0.752 (0.735-0.769) |
| Majority Voting | 6 | 0.759 (0.742-0.776) |

In conclusion, we supplemented the previously proposed clinical score generator with additional variable ranking methods and an automatic model selection strategy, thereby improving its robustness and automation. We demonstrated their effectiveness on BIDMC dataset.

URM STATEMENT

The authors acknowledge that all of us meet the URM criteria.

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

# A  METHOD DETAILS

In this section, we give an overview of AutoScore's six modules, including variable ranking, variable transformation, score derivation, model selection, score fine-tuning, and model evaluation. We then introduce five alternative variable ranking methods and an automatic model selection option to be added to AutoScore framework. Additionally, we present the model selection based on the proposed automatic approach.

## A.1  OVERVIEW OF THE AUTOSCORE FRAMEWORK

The AutoScore framework was developed to generate transparent and interpretable clinical scores automatically (Xie et al., 2020). There are six modules in the original AutoScore framework (See Figure 1). Module 1 uses random forest to rank all variables according to their variable importance. Module 2 converts continuous variables into categorical ones to facilitate the derivation of point-based scores to enhance model interpretability. Logistic regression is then applied in module 3 to produce an initial version of a risk score. In module 4, a parsimony plot is created to illustrate the relationship between model performance and complexity (measured by the number of variables), which helps determine which variables to be included in the final score. Based on users' clinical knowledge, module 5 allows the adjustment of categorical variable intervals. Module 6 assesses the performance of the developed clinical scores. In essence, the AutoScore framework combines ML-based variable ranking with statistical regression to generate interpretable clinical scores.

AutoScore package (Xie et al., 2022b) helps users simplify the six modules into three main steps. Firstly, users input the training dataset, validation dataset, and test dataset into AutoScore. Then the users are required to select models from multiple candidate models provided by AutoScore. Lastly, a clinical score and its test performance will be outputted.

It is worth noting that AutoScore uses RF as the variable ranking algorithm (module 1), which may not be the optimal ranking method in some circumstances (Strobl et al., 2007; Nicodemus, 2011). Thus, embedding additional variable ranking methods into AutoScore makes it more versatile. The other module to be improved is the manual model selection (module 4), which interrupts the automation of AutoScore. In some application circumstances where users implement many pilot studies and expect rough yet swift assessments of clinical scores, such an intermediate step is not user-friendly and it can be improved through an automatic option which also aligns with the notion of "Auto".

## A.2  ALTERNATIVE VARIABLE RANKING IN AUTOSCORE

We explore five alternative variable ranking methods for the replacement of RF (module 1). To facilitate comparison, AutoScore with RF is considered as the baseline.

### A.2.1  VARIABLE RANKING WITH STATISTICAL METHODS

Statistical tests such as the two-sample Student's t test (Student, 1908) and Wilcoxon's rank sum test (Wilcoxon, 1947) are classic methods for continuous variable ranking and the univariable p-values

derived from them are often employed to rank variables in medicine (Shri & Sriraam, 2017). Since all variables in our dataset are continuous, t test and Wilcoxon's rank sum test work well. However, t test cannot be applied to datasets that contain both continuous and categorical variables. Users should therefore avoid using this ranking method on mixed-type datasets. Variables with small P-values suggest significant differences between outcomes and are therefore assumed to have top rankings in the variable ranking module and great discriminatory power for the scoring model.

### A.2.2 Variable ranking with ML methods

RF (Breiman, 2001) and XGBoost (Chen & Guestrin, 2016) are two popular ML solutions for variable ranking. XGBoost is a scalable boosting system that provides tree-based modeling. Its variable importance is calculated through the percentage of gain, which measures the fractional contribution of each variable (Chen & Guestrin, 2016).

Artificial neural networks (ANNs) achieve remarkable performance in diverse clinical triaging tasks (Esteva et al., 2019; Zhao et al., 2021) and thus we explore ANNs-based variable ranking methods. Considering that ANNs do not have inherent variable ranking methods, we connect ANNs with SHapley Additive exPlanations (SHAP) (Lundberg & Lee, 2017) to derive variable ranking (Schultebraucks et al., 2020). As a unified post hoc model explanation tool, SHAP interprets the variable contributions to the model through SHAP values, which can be further used for variable ranking. Specifically, we calculate the sum of absolute SHAP values across all predictions to be explained and then follow the ranking rule that higher sum values indicate greater variable importance (Lundberg et al., 2018).

## A.3 Model selection in AutoScore

Based on the module 1, 2, and 3, AutoScore prepares a list of candidate clinical scores for manual selection. To suit the need of obtaining quick assessment of clinical scores in large-scale pilot studies, an option of automatic model selection would make AutoScore more user-friendly. Furthermore, we propose an ensemble method named "majority voting" which combines model selection results by various variable ranking methods to improve the robustness of AutoScore.

### A.3.1 Model selection based on individual ranking methods

The optimal model selection in AutoScore (module 4) is determined by "human observation" based on the parsimony plot, which is derived by coupling the model complexity (in terms of the number of top-ranked variables) with the prediction performance. The users then choose a cut-off point for selecting the number of variables to be included in the model. In this way, models are selected in a manner that achieves a good balance between performance and clinical relevance. We illustrate the mechanism of model selection with a six-variable problem. Using an individual ranking method, these variables are ranked, and six models are constructed using the top-ranked variables and evaluated with the validation data. For instance, the first model is created with the top one variable, and the second model with the top two variables, and so forth. When the fifth and sixth important variables are added to the models, there is only a marginal performance improvement. As a result, we select the top four variables as the basis for deriving the clinical score.

As an alternative to the "human observation" approach, we propose a heuristic "thresholding" strategy to automate the process. For example, the variables #1, #2, #4, #6, #3, and #5 are ranked in decreasing importance. With the top five variables, the AUROC reaches the highest value of 0.8. An AUROC threshold can be established by specifying a percentage (e.g., 95%) of the maximum AUROC value. During model selection, we will select the top variables that have an AUROC at or slightly above the threshold. In this six-variable demonstration, we specify an AUROC threshold equal to 95% of the maximum value, i.e., $0.8 \times 95\% = 0.76$. By applying the threshold, the top four variables that produce a model with an AUROC above and closest to the threshold is selected.

### A.3.2 Model selection based on proposed ensemble method

While individual ranking methods in conjunction with "human observation" approach provide flexibility in model selection, ensemble learning may provide an "automated" solution that could not only remove bias introduced by individual methods but also improve the robustness of the model

(Bhowan et al., 2012). As part of our study, we propose using majority voting, an ensemble learning strategy, to aggregate the outputs of individual ranking methods for model selection. Specifically, using the "thresholding" strategy described in the previous section, we can identify the most significant variables in each ranking method. For each variable, we count its occurrences among all ranking methods. The frequency of occurrences indicates the importance of the variable. According to the majority voting strategy, we kept variables that are selected by at least half of all rankings.

## B   Experimental Details

We used a dataset of 44,918 de-identified patients from the BIDMC critical care units (Johnson et al., 2016). We followed the details of the study cohort, including gender, type of admission, ethnicity, insurance, and type of intensive care unit as Xie et al. (2020). The primary outcome was mortality (8.81% of 44,918 patients). The whole dataset was randomly divided in a 60/20/20 split for model training, validation, and testing. We chose a total of 21 candidate variables for predictive modeling, including temperature, heart rate, age, respiration rate, systolic blood pressure, peripheral capillary oxygen saturation (SpO2), white blood cells, diastolic blood pressure, platelet, glucose, sodium, lactate, mean arterial pressure, potassium, bicarbonate, blood urea nitrogen, hematocrit, creatinine, hemoglobin, chloride, and anion gap. Missing variables were imputed using the population median.

The first extension of AutoScore was to integrate alternative variable ranking methods into AutoScore. As indicated by previous research (Strobl et al., 2007; Nicodemus, 2011), RF might not be the optimal ranking method in some circumstances. Our experiments also suggested that other ranking methods contributed to building better clinical scores which use fewer variables while achieving comparable discriminating ability. For example, t test-based clinical score used 5 variables and was comparable to the AUROC achieved by the RF-based clinical score with 7 variables. Such sparsity of variable numbers is of great value in a fast-paced hospital environment. Clinicians prefer clinical scores with fewer variables while holding comparable discriminative ability, because sparse variable numbers substantially eliminate clinicians' workload, and comparable discriminative ability ensures that such workload elimination will not impair model reliability. Considering there is no individual ranking method that works the best across all circumstances (Zhang et al., 2017), we presented an ensemble method named majority voting, enabling us to include all paradigms and synthesize the collective information provided by various ranking methods. Future work on AutoScore will explore more ensemble algorithms, not limited to model selection.

The other proposed extension of AutoScore is providing an automatic model selection option based on the "thresholding" computation of maximum AUROC by all candidate models. This automation aligns with the general selection rule proposed in AutoScore that an optimal clinical score should achieve a good balance between discriminating performance and model complexity. Specifically, we expect that a clinical score holds relatively accurate classification performance while applying relatively few variables. Generally, clinicians who possess domain knowledge help with this selection: They will conduct some literature review, shortlist some candidate variables and hold a panel discussion to make the final decision. However, such manual model selection requires much time and is not always user-friendly. For example, in a pilot study, users prefer quick development and evaluation of clinical scores generated by AutoScore. Thus, we provide an additional automation option in the model selection module and demonstrate its effectiveness. We hope this update could contribute to an expressway in clinical pilot studies.

Several new questions emerge in light of the discoveries in our study. First, we examined only one BIDMC dataset for demonstration. Given that model's performance might fluctuate as datasets and target tasks change, it is necessary to perform additional validations under various settings to further verify the effectiveness of current updates in AutoScore. For example, the current model selection set the threshold as 95%, and additional experiments on other datasets contribute to the credibility of such threshold. Second, the heterogeneity of clinical scenarios and the complexity of variable rankings prevented us from recommending the best individual ranking methods. In general, ensemble learning eliminates the bias introduced by individual methods and improves the robustness. Also, some ranking methods like t test are only effective in datasets consisting of continuous variables. Lastly, candidate models in AutoScore are generated in a fixed and incremental approach, which simplifies computational complexity, but the simplification might lead to the omission of the optimal model. Future studies will consider improvements in these parts.

