# OpenReview forum: "Interpretable Machine Learning-Based Risk Scoring with Individual and Ensemble Model Selection for Clinical Decision Making"
_ICLR.cc/2023/TinyPapers — Submitted to Tiny Papers @ ICLR 2023_

### Official Review · Reviewer_gnNF · 2023-03-28

**Confidence:** 4

**Summary Of Contributions:**

In this paper, the authors investigate methods to augment AutoScore, an existing framework for generating clinical scores. Specifically, the authors seek to improve and expand two modules within the framework: (1) variable ranking using random forest, and (2) model selection, transitioning from manual to automatic, in order to achieve comparable discriminating ability while utilizing fewer variables.

**Rating:**

Clear, Correct, and Reproducible (CCR): a submission which meets the reviewing criteria

**Strengths And Weaknesses:**

- **Strengths**
    - The authors use a comprehensive approach in experimenting with multiple techniques i.e. five alternative variable ranking methods utilizing both statistical and machine learning methods, as well as an ensemble ranking method via majority voting. This broad exploration of techniques helps to advance research towards a rigorous and comprehensive evaluation of the AutoScore framework.
    - The authors have shared their code, which promotes reproducibility, transparency, and collaboration within the research community.
    - The paper presents a method for moving towards an automatic option for model selection within AutoScore, by aggregating the outputs of individual ranking methods, which contrasts with the current manual model selection process.
- **Weakness**
    - The paper mentions that - “AutoScore applies random forest (RF) (Breiman, 2001) to rank variables, which is biased in some scenarios (Strobl et al., 2007; Nicodemus, 2011)”. I would encourage the authors to expand on the rationale behind their selection of the five alternative variable ranking methods, as well as any potential limitations or drawbacks of these methods in certain scenarios. This could help to provide a more comprehensive and nuanced evaluation of the overall framework.

**Suggested Changes:**

None

---

> ### Author Response · Authors · 2023-04-18
> **Reply to Reviewer gnNF**
>
> Thank you for the detailed comments and we will explore the reason behind the collapse of random forest-based variable ranking in future work.

---

### Official Review · Reviewer_zxTg · 2023-03-28

**Confidence:** 4

**Summary Of Contributions:**

This work investigates potential improvements to the “AutoScore” Framework (Xie et al. 2020) in variable selection and automatic model selection. They suggest multiple variable selection methods and propose a simple automatic model selection based on a threshold of the relative model performance to the maximum.

**Rating:**

High Potential (HP): a submission which meets the reviewing criteria and has potential to make an impact on the field

**Strengths And Weaknesses:**

Strengths:

- Figure 1 is very clear on what the paper is looking at. The paper is well written overall.
- Overall this work seems reproducible. The code has an excellent readme which clearly describes the method.

Weaknesses:

- It isn’t clear to me what curve the “AUC” metric is referring to. I’m guessing either receiver operator characteristic or precision recall. Given that an automatic score is based on thresholding this value it would be good to discuss which score this is and when you might want to threshold based on it or another score (I can see even just false positive or false negative rate being reasonable in some circumstances).
- One thing that I feel is missing in the scoring is the model complexity and not only the number of input features. A complex model built on a few features may be less desirable than a simple model built on a more features. For instance SHAP with a large MLP may be less useful overall than a random forest or linear model. This is not an issue when only comparing random forest models, but becomes an issue when comparing models of different types.

**Suggested Changes:**

- Clarify the score used and why it was chosen as well as its benefits and limitations.

- Clarify the issues with model selection (which is in general a very difficult task) that confounds selection based on the number of variables.

- Clarity of the abstract: The “… AutoScore leaves room for enhancement: variable ranking via the random forest and manual model selection. In this work, we improved them…” bit of the abstract was a bit unclear to me, but I understand it after reading the next sentence. I Would suggest: “… AutoScore leaves room for enhancement. In this work, we improved AutoScore…”

---

> ### Author Response · Authors · 2023-04-18
> **Response to Reviewer zxTg**
>
> Thank you for the detailed comments and we revised the abstract accordingly. The AUC is Area Under the Receiver Operating Characteristic Curve (AUROC). Clinical scores used simple calculations to derive the final patient risk and thus the variable number is an important indicator of model complexity (https://www.mdcalc.com/calc/1873/national-early-warning-score-news). Generally, a scoring system requiring fewer variables will release nurses' burden of data collection.

---

### Meta-Review · Area_Chair_AY7D · 2023-04-02

**Recommendation:** Invite to present
**Confidence:** 4

**Metareview:**

This work aims to improve the AutoScore framework by focusing on two aspects: variable ranking using random forest and model selection, transitioning from manual to automatic. The authors experiment with multiple techniques, including five alternative variable ranking methods and an ensemble ranking method using majority voting. Overall, the paper is well-written, clear, and reproducible, with the authors providing their code to promote transparency.

**Summary:**

The paper aims to improve the AutoScore framework by focusing on variable ranking using random forest and transitioning from manual to automatic model selection. The authors experiment with multiple techniques and provide their code for transparency.

**Comments And Feedback To The Authors:**

* Clarify the "AUC" metric, its role in the automatic model selection process, and discuss the benefits and limitations of using it.
* Expand on the rationale behind the selection of the five alternative variable ranking methods, their potential limitations, and drawbacks in certain scenarios.
* Consider refining the abstract for improved clarity, as suggested by Reviewer zxTg.


**Reason For Not Giving A Higher Recommendation:**

The paper has potential, but some issues need to be addressed to improve its clarity and significance. The authors should clarify the "AUC" metric, its role in the automatic model selection process, and discuss the benefits and limitations of using it. Additionally, the rationale behind the selection of alternative variable ranking methods and their potential limitations should be expanded upon to provide a more comprehensive evaluation of the overall framework.


**Reason For Not Giving A Lower Recommendation:**

The paper presents a comprehensive approach to augmenting the AutoScore framework. The authors explore multiple techniques, provide their code for transparency, and effectively communicate their research. Despite some limitations in the discussion, the paper has the potential to make an impact on the field.

---

> ### Author Response · Authors · 2023-04-18
> **Response to Area Chair AY7D**
>
> Thanks for the suggestion. The AUC is Area Under the Receiver Operating Characteristic Curve (AUROC) and we supplemented the updated version with variable selection method in Appendix A.3 (Model Selection in AutoScore) .

---

### Decision · Program_Chairs · 2023-04-09

Invite to present